# Parenting Stress, Maternal Self-Efficacy and Confidence in Caretaking in a Sample of Mothers with Newborns (0–1 Month)

**DOI:** 10.3390/ijerph19159651

**Published:** 2022-08-05

**Authors:** Giacomo Tognasso, Laura Gorla, Carolina Ambrosini, Federica Figurella, Pietro De Carli, Laura Parolin, Diego Sarracino, Alessandra Santona

**Affiliations:** 1Department of Psychology, University of Milano-Bicocca, 20126 Milano, Italy; 2Department of Developmental Psychology and Socialisation, University of Padua, 35131 Padova, Italy

**Keywords:** attachment, maternal confidence, self-efficacy, newborn, caretaking

## Abstract

A mother’s responses to her newborn and her confidence in the child’s caretaking depend on her attachment security, general parental stress, and perceived self-efficacy. However, few studies have analyzed maternal confidence in caretaking and how it is influenced by some mothers’ characteristics. We aimed to examine the association between maternal adult attachment and confidence in a child’s caretaking and to understand whether this relationship was mediated by parenting stress and maternal self-efficacy. The sample consisted of 96 mothers with a mean age of 33 years with newborn children aged between 3 and 30 days. The instruments used were the Experiences in Close Relationships-Revised (ECR-R), the Mother and Baby Scale (MABS), the Parenting Stress Index Short Form (PSI-SF), and the Maternal Self-Efficacy Questionnaire (MEQ). The results showed a positive association between attachment avoidance and lack of confidence in caretaking, and this association was mediated by parenting stress. Conversely, attachment anxiety appeared not to influence confidence in caretaking, and maternal self-efficacy did not appear to mediate the relationship between attachment and confidence in the caretaking of infants. Our results could guide new research in studying confidence in caretaking and enable healthcare professionals to recognize at-risk situations early from the first month after childbirth.

## 1. Introduction

Transition to parenthood is a major developmental period that includes practical and psychological challenges for new parents [1,2,3,4,5]. First, they must understand and act according to the needs of the newborn, modifying their daily routines to deal with new tasks and greater responsibilities [6]. This implies a change within the couple and could cause a lower relationship satisfaction since parents experience less time for couple communication or lower quality and frequency of time for the couple itself [7].

The important role that a couple has in the well-being of the new family has been highlighted by some studies, which report a strong association between couple interaction and parenting behavior [8,9], and find that satisfying couple relationships increase responsiveness toward childrearing and are associated with confidence and competent parenting [10].

Psychological challenges connected to the transition to parenthood are mainly connected to the development of different identities (self, parent, and partner) which can be difficult to handle, and which can require some time to become comfortable with [3,11]. Indeed, transition to parenthood is a reorganization process that involves both members of the couple, especially the mother [3,12,13], who is asked to rearrange both internal and external dynamics in terms of differentiating from the family of origin and re-elaborating the position as a daughter to move towards the development of an idea of herself as mothers [3].

As the parenting role assumes priority, mothers often postpone their own needs to respond promptly to their child’s needs, which can affect the mother’s well-being and her relationship with her partner [14,15]. For this practical and psychological reason, new mothers often report fatigue and exhaustion, psychological distress, and struggling with too many new tasks and responsibilities [4].

As briefly mentioned above, several factors are involved in the transition process of becoming a mother; in this paper we will focus on adult attachment, parenting stress, maternal self-efficacy, and maternal confidence in caretaking [1,16,17,18].

### 1.1. Adult Attachment

For attachment theory, adults’ style of parenting is widely affected by the Internal Working Models [19], which are internalized representations of the self, others, and self–other relationships, based on childhood experiences with attachment figures [20]. Adult attachment styles can fit into secure or insecure patterns [19]. The secure pattern is based on internalizing a positive self-model and positive conception of others, whereas the insecure pattern is defined as a two-dimensional model of attachment-related anxiety and attachment-related avoidance [21]. Attachment anxiety is the fear of not being loved by a partner or of being left by the partner, while attachment avoidance concerns unease with dependency and intimacy [22].

Adult romantic attachment is linked to parental caregiving and parenting expectations, experiences, interaction, and behaviors [23,24,25]. In particular, attachment avoidance and anxiety are associated with more negative expectations about parenting, including uncertainty regarding parenting ability, expectations of being easily aggravated by and relating poorly to children, and having a less warm, more strict parenting style [26]. Moreover, high avoidance was associated with an expectation of less overall satisfaction in parenting [26,27,28]. Attachment anxiety has also been linked to difficulties in understanding some child’s requests and in supporting a child’s exploratory behaviors, whereas attachment avoidance has been reported to be positively associated with distance in caregiving interactions and negatively associated with maternal sensitivity, especially in conditions of psychological distress [29,30].

### 1.2. Parenting Stress

Another key factor that affects mother–infant bonding is the stress connected to the transition to parenthood. In particular, parenting stress is associated with child, parent, and contextual factors [31,32]. 

Child factors related to a mother’s parental stress include the child’s temperament (which can be perceived as difficult or not), the mother’s expectations of the child, and the mother’s perceptions of difficulties in adjusting to her parenting role. Contextual factors associated with parenting stress include the relationship with the partner, the social support available, the mother’s health, and the restrictions caused by the parenting role. Finally, parent factors linked with parenting stress include the extent to which a mother is emotionally and physically available to her child, her sense of competence in the parenting role, her intrinsic investment in the parenting role, but also her temperament, psychological functioning, and attachment style [29,33,34].

The level of a mother’s parental stress is influenced by contingent interactions between all these factors; in particular, there is a strong interaction between parenting stress and the mother’s attachment style within the couple [16,18,34,35]. Under stressful conditions, individuals with insecure attachment may engage in anxious/uncollaborative or avoidant/disengaged behaviors following their underlying working model. Moreover, insecure attachment styles could be associated with less sensitive parenting, especially if there are high levels of stress [29].

Several studies have underlined that mothers with insecure-avoidant attachment styles are more likely to disengage emotionally from interaction with the child and to show fewer sensitive styles of caregiving. However, attachment-based differences in parenting were dependent on the mother experiencing elevated levels of psychological distress, not only parenting stress. Likewise, the relationship between the adult attachment style and bonding was mediated by parenting stress: particularly, higher scores on attachment avoidance and anxiety were related to increased stress, which was related to decreased quality of bonding [16].

### 1.3. Mother Self-Efficacy (MSE)

Going into more specific maternal factors that play a role in the development of parenting behaviors, one important aspect that should be considered is maternal self-efficacy [34]. Self-efficacy is the individual perception and trust in one’s own ability to perform a particular behavior successfully [36]. The concept of self-efficacy could be linked to the parenting skills and tasks required of the mother and her own beliefs regarding the successful enactment of these behaviors. Bandura [36] emphasized that mothers must believe that their actions will have the desired outcome and have confidence in their ability to perform specific behaviors or skills. If a woman feels that she can take good care of her baby, her self-esteem will increase and she will be able to show affectionate reactions to her baby’s requests, building positive interactions with the baby. On the other hand, in a situation of less self-confidence in caring for the baby, performing the maternal role could be very difficult [37,38]. Several authors registered an increase in maternal self-confidence during the months postpartum [13,39,40]; moreover, multiparas were found to have higher levels of MSE than primiparas, and levels of MSE increased over time for both primiparas and multiparas [34,41,42] supporting Bandura’s theory that childcare experience enhances MSE. 

The MSE is essential for mothers’ satisfaction with parenting and results in a protective factor against postpartum depression, stress, and anxiety [13,17,41,43,44,45]. Indeed, maternal stress, anxiety, and depression are negatively associated with a high level of MSE [13,45,46]. However, MSE at one month was also associated with marital positivity (love and maintenance), which supports Bandura’s theory that social-marital support influences self-efficacy through processes involving social persuasion and verbal encouragement [34,36].

### 1.4. Maternal Confidence in Caretaking (MCC)

Strictly connected to maternal self-efficacy, maternal confidence in caretaking can be described as a task level of MSE which affects the moment when the mother responds to the primary child’s needs such as breastfeeding, bathing, comfort, and falling asleep [47]. MCC is essential because mothers’ ability to care for their infants can impact the physical, cognitive, and socio-emotional development of their newborns [48,49,50].

There are few studies specifically focused on confidence in infant care practices for new mothers in the postnatal period; but the literature reports that mothers’ lack of confidence with their infants in the early postnatal period may negatively influence their experience of motherhood and hence their ability to adequately care for the infant [34] Leahy-Warren [34] found that, as for MSE, MCC in infant care behaviors commonly performed (such as holding the infant) is higher than in those that are less routinely performed (such as recognizing croup). Furthermore, appraisal and informational support positively influence first-time mothers’ confidence in infant care practices.

MCC, MSE, attachment, and parenting stress play an essential role in building the relationship between the new mother and her child. For this reason, some studies focus on the mediation effect between these variables [16,44,50].

For instance, Nordahl [16] found that the relationship between the adult attachment style and mother–infant bonding is mediated by parenting stress; higher scores on attachment avoidance and anxiety were related to a high level of stress and were connected to a decrease in the quality of bonding. 

Moreover, a longitudinal study [51] examined the changes in both mothers’ and fathers’ parental adjustment over time and the mediating role of common dyadic coping (the way that parents interact about parenting questions) on the associations between anxious and avoidant attachment, parenting stress, and parental confidence. They found that, while perceived parenting stress declined throughout the first year after the child’s birth, parental confidence increased. Moreover, parents with avoidant attachment were more likely to not present a common dyadic coping from pregnancy to six weeks postpartum, increasing their partners’ parenting stress (especially if the partners were mothers), and decreasing their partners’ parental confidence.

Regarding MCC, a study [52] reported the importance of maternal confidence and maternal competence in mediating maternal parenting stress. 

Nevertheless, to the best of our knowledge, a mediation model that integrates adult attachment, parenting stress, MSE, and MCC has yet to be done in the literature.

Although the interaction between adult attachment, parenting stress, and MSE has been widely studied, there is a paucity of research about MCC. In particular, the few studies on this theme have focused only on the months following childbirth, without considering the period immediately after the delivery. 

### 1.5. Aims of the Study

The present study aims to explore the complex relationship between the mother’s attachment, parenting stress, self-efficacy, and maternal confidence in caretaking in order to shed light on the mechanisms through which adult attachment influences the mother’s confidence in caretaking, and whether the levels of parenting stress and maternal self-efficacy could mediate between attachment and confidence in caretaking.

To the best of our knowledge, the results could fill a gap in the literature regarding maternal confidence in caretaking, an aspect that has not yet been studied in connection to adult attachment, self-efficacy, and parenting stress simultaneously.

Another novel aspect of the current study is that we analyzed these aspects focusing only on the first month postpartum, aiming to understand how maternal confidence in caretaking is influenced by mothers’ attachment style and parenting stress experienced in the first period of being a mother. 

According to the literature above reported, we expected that:

(a)The relationship between adult attachment (evaluated in both anxiety and avoidance dimensions) and lack of confidence in a child’s caretaking is mediated by the mothers’ self-efficacy and parenting stress.(b)In particular, a high level of anxiety and avoidance predicts a high level of lack of confidence, and this relationship is mediated by high levels of parenting stress.(c)A high level of maternal self-efficacy positively mediates the relationship between adult attachment and lack of confidence in a child’s caretaking, acting as a protective factor in this relationship.

## 2. Materials and Methods

### 2.1. Participants and Procedure

Our sample was composed of 96 Italian first-time mothers aged between 25 and 48 years (*M* = 32.9; *DS* = 5.2), who had children aged between 3 and 30 days (*M* = 10.7; *DS* = 4.9). 

All the mothers were involved in a relationship and most of them (43.8%, *n =* 42) were receiving daily support from their partner, followed by 42.7% (*n* = 41) helped by both the partner and the family, 12.5% (*n* = 12) by the only family and only one participant (1%) with no support or help. Mothers rated the support they were having as high (67.7%, *n* = 65) and moderate (31.3%, *n* = 30), with only one participant who did not answer (1%, *n* = 1).

Regarding the child’s nursing, most of the mothers (71.9%; *n* = 69) nursed their children with a mix of natural breastmilk and artificial milk, while 27.1% (*n* = 26) only breastfed with breastmilk and only 1% (*n* = 1) used exclusively the artificial milk. All the children were healthy full-term infants.

We collected data in collaboration with two Italian hospitals (San Raffaele and Buzzi hospitals) during the usual check-up made by doctors a few days after the delivery.

We asked to participate in the current study only first-time mothers who were described by the doctor as having had a safe, full term delivery.

In particular, a research assistant waited for the mothers to end the visit and asked them to fill in a questionnaire that collected information about the mother and child’s age, the child’s delivery and breastfeeding, and the mothers’ social support, and asked them to compile all the standardized instruments. The research assistant explained the research to the mothers and was present at all times during the questionnaires’ fulfillment. The mothers who decided to participate in the study filled the instruments in a private room without being interrupted. 104 first time mothers were invited to participate in the current study and 96 of them agreed, granting us a response rate of 92%.

Data were collected between January and September 2019, before the outbreak of COVID-19, so the procedure for giving birth in Italian hospitals was not affected by the pandemic situation.

We followed the provisions of Italian law 196/2003 to collect the participants’ consent to be interviewed and to participate in the research. Before starting, the participants read a brief explanation about the content and purpose of the study. The research project was previously approved by the Ethics Committee of the Psychology Department of Milano-Bicocca University.

### 2.2. Measures

*Mother’s Attachment*: We used the Experiences in Close Relationships Scale-Revised (ECR-R) [22] in the Italian version [53] is a 36-item self-report instrument that assesses feelings and behavior related to attachment in romantic relationships. Items are rated from 1 (“strongly disagree”) to 7 (“strongly agree”) with higher scores associated with higher endorsement of the construct. The instrument allows classifying the romantic attachment by clustering the items in two dimensions: *Avoidance of intimacy* and *Anxiety about abandonment.* The first one measures the level of preoccupation related to sharing emotional nearness (i.e., “I prefer to not show my partner how I feel deep down”), while the second one measures the preoccupation with the relationship or the need for intimacy (i.e., “I worry about being alone”). In our sample, the omega coefficient was 0.61 for the avoidance dimension and 0.37 for the anxiety, showing moderate reliability that should be considered in interpreting the scores from this scale.

*Maternal confidence in caretaking*: We used the Mother and Baby Scale (MABS) [54] to evaluate the mother’s experience in childcare. The MBAS is a self-report 36-item questionnaire composed of two parts: the first one focuses on the maternal representation of the baby (i.e., *baby’s irritability, infant perceived as difficult*), while the second one focuses on the mother’s feeling of competency (i.e., *lack of self-confidence, lack of competence in childcare and lack of confidence about feeding*). 

The MBAS is formed by five subscales regarding the maternal representation of the baby (activity level, reactivity, discomfort, irritability, and infant perceived as difficult) and three subscales focused on the mother’s feeling of competency: lack of confidence in infant care (LCC), lack of self-confidence, and lack of confidence about feeding. In the current study, we focused only on the mother’s feeling of lacking confidence in infant care. Parents are asked to score the various statements with a score between 0 (not at all) and 5 (very much/often), with a score range for the lack of confidence subscale from 0 to 65, with higher scores on the lack of confidence scale suggesting that the mother is less confident in taking care of the baby. We calculated the omega coefficient for the LCC subscale and found that it was 0.87, showing a high level of internal consistency.

*Parental Stress*: We used the Parenting Stress Index- short form (PSI-SF) [31] to measure parenting stress perceived by the caregivers in the parent-child system and to identify the presence of some problems in parent-child interactions. The PSI-SF is a 36-item self-report questionnaire with three subscales: *Parental Distress* (PD-SF) is the level of stress experienced by the parents towards the parenting role; *Parent-Child Dysfunctional Interaction* (PCDI-SF) assesses the parent’s perception of the child as a subject not meeting expectations and not satisficing in the relationships; and *Difficult Child* (DC-SF) surveys the parent’s view of the child’s temperament, demandingness, behaviors, and compliance. Each subscale is formed by 12 items rated from 1 (“strongly disagree”) to 5 (“strongly agree”). Every subscale’s scores range from 12 to 60 with a total score of the instrument from 36 to 180. High scores indicate great levels of stress. In our sample, the omega coefficient was 0.93 for the total score, showing satisfactory reliability.

*Maternal self-efficacy*: We used the maternal self-efficacy Questionnaire (MEQ) [55] to evaluate mothers’ self-efficacy beliefs regarding specific areas of infant care (e.g., *soothing the child, maintaining joint attention, and interaction with the child, and performing daily routines*). The questionnaire is formed by ten 4-points items rated from 1 (“not good at all”) to 4 (“very good”). Scores are summed to derive a total score and the potential range of scores is from 9 to 36, with higher scores indicative of stronger self-efficacy beliefs. In our sample, the omega coefficient was 0.92 showing a high level of internal consistency.

### 2.3. Analysis Plan

Firstly, we conducted descriptive statistics and bivariate correlations among research variables. We also performed t-tests to make a comparison between our sample and comparable groups taken from previous studies and literature. Secondly, to explore whether maternal efficacy and parenting stress mediate the association between attachment dimensions and lack of confidence in caretaking, we conducted path analyses for anxiety and avoidance separately. Multiple regression analyses were used to test whether there is a significant mediated effect, while a path analysis was subsequently conducted to visualize a general pattern of associations between all of the variables. Analyses were run using the statistical software IBM SPSS version 27.

## 3. Results

### 3.1. Descriptive Analyses and Bivariate Correlations

Table 1 reports the means, standard deviations, and correlations for the study variables. The ECR anxiety was not significantly related to any of the other variables, while the ECR avoidance significantly correlated to lack of confidence (LCC). In addition, the lack of confidence was significantly correlated both to the parenting stress index and maternal self-efficacy. We also discovered a significant correlation between maternal self-efficacy and parenting stress.

We also performed unpaired two samples t-tests comparing our sample to samples taken from the literature [56,57,58,59]. Table 2 shows that our sample had higher scores on both anxiety and avoidance attachment and parenting stress than the comparison groups while obtaining fewer scores in maternal self-efficacy. This means that our mothers were more anxious and avoidant and perceived a higher level of parenting stress and a lower maternal self-efficacy compared to the other sample. By contrast, the lack of confidence in caretaking was not statistically different from the normative sample. However, the t-test effect size shows that our sample differs from the normative sample especially in the avoidant dimension, with a limited effect size for all the other dimensions.

### 3.2. Mediation Model

To address Hp 1, we firstly conducted a series of regression analyses to assess the possible mediation effect of maternal self-efficacy and parenting stress in the relationships between adult attachment and the mother’s lack of confidence in caretaking (see Figure 1 for hypothetical model). 

In our sample, the results show that only the attachment’s dimensions of avoidance directly affect the maternal lack of confidence, while the anxiety did not have an impact on the lack of confidence in caretaking (Avoidance: F (1, 93) =4.032, *p* = 0.048; Anxiety: F (1, 93) = 0.008, *p* = 0.930). Moreover, the anxiety dimension seemed to not directly influence maternal efficacy (F (1, 92) = 2.782, *p* = 0.099) and parenting stress (F (1, 92) = 0.769, *p* = 0.383). On the other hand, the avoidant attachment had a significant effect on parenting stress (F (1, 92) = 7.298, *p* = 0.008), but did not have a direct impact on maternal self-efficacy (F (1, 92) = 0.071, *p* = 0.791).

Since the anxiety in attachment did not appear to predict both the lack of confidence and the hypothesized mediators of maternal self-efficacy and parenting stress, we elected to exclude the anxious attachment from the mediation model, focusing only on the attachment’s dimension of avoidance. 

If we consider specifically the indirect effects (see Table 3), it appears that the level of avoidant attachment positively influences parenting stress, which in turn has a positive effect on the level of confidence in childcare. This means that mothers with avoidant attachment are likely to experience a high level of parenting stress, which in turn increases the perceived lack of confidence in the child’s caretaking.

The avoidant attachment has a negative impact on maternal self-efficacy, which in turn impacts the lack of confidence in caretaking. Indeed, mothers with higher levels of avoidant attachment seem to have less maternal self-efficacy and experience a greater lack of confidence.

To address Hp2 and Hp3, we performed two different multivariate regression models depending on adult attachment dimensions of avoidance and lack of confidence in caretaking. 

The mediator was parenting stress in the first model and maternal self-efficacy in the second model (see Figure 2a and Figure 2b, respectively).

In the first model (Hp2), we first ran a regression with the avoidance dimension of attachment as a predictor and LCC as the outcome. The results showed that the relationship was statistically significant, so we conducted a regression with avoidance as a predictor and the total score of parenting stress as the outcome, finding a significant relationship (F (1, 92) = 7.298, *p* = 0.008). As observed previously, a higher level of avoidant attachment predicted a higher level of parenting stress.

Finally, we ran a regression with attachment avoidance and parenting stress as the predictors and lack of confidence in caretaking as an outcome. Our results showed a significant relationship (F (2, 91) = 11.856, *p* < 0.001) meaning that the level of avoidant attachment positively influences the levels of parenting stress, which in turn influences the lack of confidence. This means that mothers with an avoidant attachment are likely to experience a higher level of parenting stress and parenting stress tends to increase the perception of lacking confidence in a child’s caretaking. To test the model significance, we used the Sobel test (test = 2.35, *p* = 0.01) which confirms that parenting stress could mediate the relationship between avoidant attachment and lack of confidence.

To verify Hp3, we first tested the relationship between avoidance and maternal efficacy and found that it was not statistically significant (F (1, 92) = 0.071, *p* = 0.791). Then, we conducted a second regression using maternal self-efficacy as the predictor and the lack of confidence as the outcome, finding a relationship statistically significant (F (1, 92) = 32.05, *p* < 0.001): a higher level of maternal self-efficacy appeared to decrease the feeling of not being confident in child’s caretaking. 

After this, we executed a regression with avoidance as the predictor, lack of confidence as the outcome, and maternal self-efficacy as the mediator and we discovered that the mediation was statistically significant (F (2, 91) = 18.936, *p* < 0.001). To test the model significance, we used the Sobel test (test = 0.26, *p* = 0.79) which confirmed that maternal self-efficacy appeared not to mediate the relationship fully or partially between avoidant attachment and lack of confidence in caretaking.

## 4. Discussion

We wanted to explore maternal confidence in caretaking, a theme that has not been widely studied in the literature but is strongly present in the process of becoming a mother [49,50]. In particular, the current study aimed to deeply analyze the relationship between the mother’s attachment and her feeling of confidence in a child’s caretaking and to understand whether this relationship is mediated by parenting stress and maternal self-efficacy.

As the descriptive analysis show, our sample was composed of mothers with a high level of both anxiety and avoidance attachment, high parenting stress, and low maternal self-efficacy.

These results are in line with the literature that highlights the challenges and difficulties present in the first period after the child’s birth [5,34]. Becoming a mother could be a very stressful life event since involves a transition from a well-known reality to another one, characterized by new goals, behaviors, responsibilities, and a new concept of self [1,14]. When a child is born, mothers are asked to change their habits, understand the child’s signals, respond to his needs [32,47], and develop a maternal role that could take some time to be learned and internalized [18,55]. The transition to motherhood has been theorized as connected to mothers’ fatigue, physical exhaustion, psychological distress, and less perception of their mothers’ capability [4]. All these aspects could be an explanation for the high levels of stress that the mothers of the current study experienced.

It is interesting to note that in our sample the stress widely present in the first days of motherhood did not appear to be decreased by the support received. Indeed, although the mothers reported to be in a significant relationship and to be supported in their daily life with their newborns by their partner and family, they experienced a high level of parenting stress. This aspect further reveals the strong impact the child’s birth has on the psychological well-being of mothers. We could also hypothesize that social support did not reduce the parenting stress in the first days after the delivery, but it takes a longer time to be a protective factor for the well-being of mothers. Indeed, the first days of a mother and child relationship are a delicate period, characterized by daily discoveries and learnings.

Moreover, our results show that mothers also perceived themselves as less competent in the new role than the normative sample. This aspect could be explained because they are passing through the period immediately close to childbirth when physical recovery is still in progress and the mother and the newborn are still knowing each other [3].

Consistently, the mothers are characterized by high levels of avoidant and anxious attachment, dimensions associated with more negative expectations about parenting, including uncertainty about parenting ability [24]. These results are in line with the literature that highlights how having a baby activates the mother’s attachment system [19,24,32], and how the attachment is linked to parental caregiving [23,24,25].

Interestingly, our sample did not differ from the normative one regarding the lack of confidence in caretaking. This result could be explained by the development of maternal confidence in caretaking during the first year of the mother and child relationship. Indeed, several studies confirmed that maternal self-confidence develops and increases during the postpartum period [13,39,40]. As we tested the mothers immediately after the child’s birth, we could expect an increase in these low levels over time [34,41,42], supporting Bandura’s theory that childcare experience enhances a mother’s self-efficacy and, consequently, a mother’s confidence in caretaking.

Furthermore, the mothers of our sample are young adults (with a mean age of 32.9 years), and primiparas, at the first experience of taking care of a newborn. This means also that they are experiencing for the first time the process of transition to parenthood [4]. 

Regarding the mediation model hypothesized, our results showed that only the attachment’s dimension of avoidance directly affects the maternal lack of confidence, while the anxiety did not have an impact on this factor; moreover, attachment anxiety does not appear to directly influence maternal efficacy and parenting stress. 

These results are partially in line with the literature: indeed, some studies [15] showed that both attachment avoidance and anxiety correlated significantly and positively with the level of parenting stress and the feeling of being competent in the parenting role; however, in general, findings of relations between adult attachment, parenting stress, and mother’s feelings towards the child are more consistent for avoidance than for anxiety [26]. 

These differences between the two dimensions of an insecure pattern of attachment could be explained considering the anxiety’s characteristics: in highly demanding situations, such as the postpartum period, the fear of not being good enough and the anxious activation could be seen as a protective factor. More specifically, anxiety in attachment could be managed by the new mother by being more alert and heedful to the child’s signals and needs, allowing her to respond speedily to the child and act proactively [60,61].

Our results suggest that avoidant attachment has an impact on the perception of a lack of confidence in the child’s caretaking, and this relationship is mediated by the mother’s parenting stress. These data are in line with the literature focused on the role of parenting stress [16,32]. In particular, Nordahl and colleagues [16] proved that the stress in the parenting role mediated the relationship between attachment avoidance and bonding. Our results regarding the relationship between avoidant attachment, lack of confidence, and parenting stress are in line with the literature that shows how, under stressful conditions, individuals with insecure attachment may engage in uncollaborative or avoidant and disengaged behaviors following their underlying working model, aspects associated with less sensitive parenting, especially if there are high levels of stress [28].

We also found that a higher level of maternal self-efficacy decreased the feeling of not being confident in a child’s caretaking, but maternal self-efficacy did not appear to mediate the relationship between avoidant attachment and confidence in the caretaking of infants. 

The first result could be easily cleared up by Bandura’s theory: if a woman feels that she can take good care of her baby, her self-esteem will increase, and she will be able to respond appropriately to her baby’s requests, building positive interactions with the baby [36]. On the other hand, in a situation of less self-confidence in caring for the baby, performing the maternal role could be very difficult [37,38], with the consequence of lower confidence in caretaking.

The most surprising result of this model is that the relationship between avoidance and maternal efficacy is not statistically significant. Zietlow [46] found that attachment insecurity turned out to be the strongest predictor of maternal self-confidence at pre-school age, and also other research reported negative associations between attachment insecurity and self-confidence [25]. However, these studies were not focused on the early postpartum, and research has highlighted that maternal self-efficacy evolves and changes over time [13,39,40]. In the current study, we did not follow the development of maternal self-efficacy, but we administered the instruments only one time, immediately after the delivery; this could have affected the results and could be seen as a possible explanation for the absence of a mediation effect of maternal self-efficacy in the relationship between attachment and lack of confidence. 

Since time plays an essential role in developing maternal self-efficacy, it could be possible that its weight in the mentioned mediation model will change over time. For these reasons, other studies will be required to test the mediation model and to deepen our knowledge regarding the relationship between maternal attachment, lack of confidence, and self-efficacy.

The current findings advance our knowledge about early caregiving behaviors. Specifically, they help to connect the attachment process, the perceived parenting stress, and maternal self-efficacy and highlight that an avoidant maternal attachment could be a possible source of risk for early parenting behaviors and feelings, especially in conjunction with parental psychological distress.

These aspects become more relevant if we consider that the parental psychological distress could be higher in the first days after the child’s birth, so it is extremely important to focus the attention on the first month of the child’s life. The study’s interest in the relationship between parenting stress, maternal self-efficacy, adult attachment, and perception of confidence in a child’s caretaking during the first four weeks after the delivery could be an added value and provides the basis for further research in the field. 

Indeed, our results could be useful both for researchers who study the postpartum period and for clinicians who work with mothers who are facing the challenge of becoming a mother. The results might be useful in the development of psychoeducational and perinatal psychological interventions, particularly by including the assessment of lack of confidence in caretaking in the perinatal period. In addition, the above results may be used to identify some specific mothers’ needs from the beginning and to promote actions to prevent difficulties and increase the mother’s and child’s well-being.

However, this study has some limitations. First, it has a cross-sectional design which does not allow us to draw a firm conclusion regarding the causal direction of the associations found. Second, we did not explore all the aspects and the subscales that characterize parenting stress and maternal self-efficacy, but we decided to focus only on the global indices to have a global idea of these dimensions in our sample. 

Third, we did not consider the role that some child’s characteristics (i.e., *irritability, physical conditions, and alertness*) could have on the mother’s feelings of confidence as a mother, but we concentrated only on maternal aspects, specifically on the lack of confidence.

Finally, we studied only mothers of full term babies, so in the future, it would be useful to replicate the analyses of this study with mothers with pre or post-term babies, to extend the results to a larger sample. 

Moreover, it would be interesting to understand whether the relationship between maternal attachment, lack of confidence in caretaking, parenting stress, and maternal self-efficacy would change thought the first six or twelve months after the child’s birth. In the future, it would be useful to evaluate the dimensions explored in the current study by creating a follow-up procedure, to increase our knowledge about this theme. 

## 5. Conclusions

In the current study, we explored the complex relationship between mothers’ attachment, parenting stress, maternal self-efficacy, and confidence in caretaking, aiming to understand whether adult attachment influences the mother’s confidence in caretaking and if parenting stress and maternal self-efficacy could mediate in this relationship.

We focused on the first month after the child’s birth which represents a specific moment in the mothers’ lives. Mothers go through several changes and challenges during the first weeks after the child is born because they need to change their habits, develop a new vision of self, and learn how to understand and respond to the child’s signals.

All these aspects could explain why mothers in our sample were experiencing a high level of stress and perceived a low level of maternal self-efficacy. They were also more anxious and avoidant than the comparison groups, an aspect that should be considered as becoming a mother reactivating the attachment system.

To the best of our knowledge, this is one of the few studies regarding the concept of confidence in caretaking and how it is influenced by adult attachment, parenting stress, and maternal self-efficacy. For this reason, this study sheds a light on a novel path in research and could help both the researchers and the clinical workers to better understand the specificity of the first period after the delivery. Indeed, the new aspects highlighted by the current research could be used in working with mothers who are dealing with a new concept of themselves and a new role as mothers.

The most important result we found was that the avoidant attachment appeared to decrease maternal confidence in caretaking and that this relationship could be mediated by the level of parenting stress: women who have an avoidant attachment are likely to perceive themselves as lacking confidence in caretaking, especially if they are dealing with a high level of parenting stress. These results are in line with the literature that emphasizes the negative effects of parenting stress and underlines that parenting stress must be considered a risk factor for the mother-baby relationship and maternal well-being. 

Surprisingly, the anxiety attachment appeared not to impact both the maternal confidence in caretaking, the parenting stress, and maternal self-efficacy.

Another interesting aspect we discovered was that the avoidant attachment was not mediated by maternal self-efficacy in the relationship between attachment and maternal self-confidence in caretaking. 

The knowledge developed and the concepts presented in this study may help both researchers and clinical psychologists in their work with mothers who are dealing with the period immediately after the delivery. These results highlight the significant relationship between mothers’ avoidant attachment and mothers’ feeling of lacking competence in caretaking.

Future research should focus on this relationship and study aspects of it in greater detail, exploring how the relationship between maternal attachment and lack of confidence in caretaking changes not only in the first month after the delivery but throughout the first year from the child’s birth.

## Figures and Tables

**Figure 1 ijerph-19-09651-f001:**
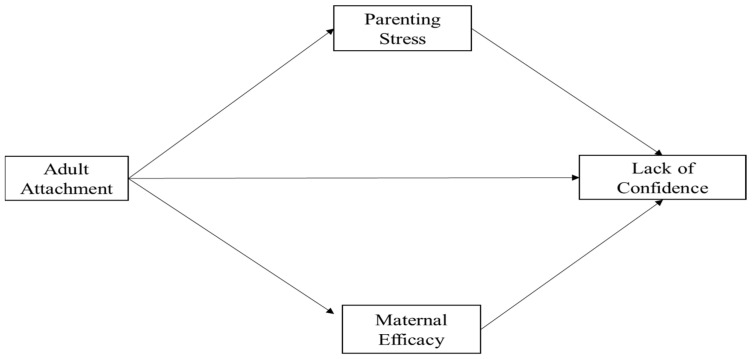
Hypothetical model. Outcome: lack of confidence in caretaking (MBAS); mediator: Maternal Self-Efficacy (MEQ) and Parenting Stress (PSI-SF); predictor: Adult Attachment-Anxiety and Avoidance Dimensions (ECR-R).

**Figure 2 ijerph-19-09651-f002:**
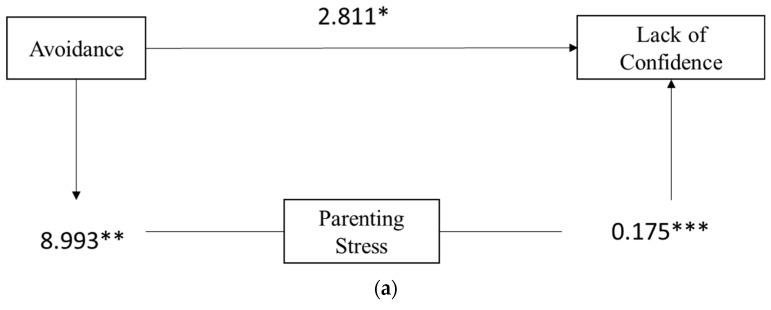
(**a**) Hp2. Path analysis with lack of confidence in caretaking as the outcome, parenting stress as the mediator, and adult attachment (avoidance dimension) as the predictor. * *p* < 0.05; ** *p* < 0.01; *** *p* < 0.001. (**b**) Hp3. Path analysis with lack of confidence in caretaking as the outcome, maternal efficacy as the mediator, and adult attachment (avoidance dimensions) as the predictor. * *p* < 0.05; ** *p* < 0.01; *** *p* < 0.001.

**Table 1 ijerph-19-09651-t001:** Descriptive statistics and Spearman’s correlations between the study’s variables.

	Mean	SD	Anxiety	Avoidance	LCC	MEQ
Anxiety	3.44	0.50				
Avoidance	4.71	0.63	−0.450 **			
LCC	29.79	8.82	0.009	0.204 *		
MEQ	19.59	4.45	0.171	0.028	−0.508 **	
PSI-SF	145.97	21.25	−0.024	0.150	0.446 **	−0.387 **

*n* = 96; * *p* < 0.05, ** *p* < 0.01 ECR: Experiences in Close Relationships, LCC: MBAS-lack of confidence; PSI-SF total score: Parenting Stress Index- total score; MEQ: Maternal Efficacy Questionnaire.

**Table 2 ijerph-19-09651-t002:** Table of comparison.

	Our Sample	Comparison Group *			
	Mean (SD)	Mean (SD)	t	d	*p*
Anxiety	3.4 (.50)	1.75 (1.01)	15.4	1.63	<0.001
Avoidance	4.7 (.63)	2.03 (1.16)	21.63	2.30	<0.001
Lack of confidence	29.7 (8.80)	26.5 (14.5)	1.57	0.26	0.11
Parenting Stress	145.9 (21.2)	124.96 (21.81)	5.71	0.97	<0.001
Maternal Self-efficacy	19.5 (4.4)	34.31 (3.63)	3.42	0.94	<0.001

* Comparison groups: Sibley, Fischer, Liu (2005) for the attachment dimensions; *n* = 82 [56]; Wolke e St James Robert (1987) for the lack of confidence; *n* = 40 [57]; Epifanio et al. (2015) for parenting stress; *n* = 53 [58]; Fulton et al. (2012) for maternal self-efficacy; *n* = 108 [59].

**Table 3 ijerph-19-09651-t003:** Defined parameters.

	Lack of Confidence in Caretaking (LCC) as Outcome *		
	ß	*p*	R-Square
**Parenting Stress as the mediator**			
Direct effect	1.288		
Total effect	0.0175	<0.001	0.207
**Maternal Self-Efficacy as the mediator**			
Direct effect	−2.602		
Total effect	−0.999	<0.001	0.294

* Avoidant attachment as the independent variable, parenting stress, and maternal self-efficacy as the mediators.

## Data Availability

The data presented in this study are available on request from the corresponding author.

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
