# Peer review of "Parenting Stress, Maternal Self-Efficacy and Confidence in Caretaking in a Sample of Mothers with Newborns (0–1 Month)"

_ijerph, 2022, doi:10.3390/ijerph19159651_

Round 1
Reviewer 1 Report
Thank you for the opportunity to review your manuscript. It sheds light on some really important variables that are impacting a mother's caretaking of her infant. I have a few comments to make some improvements in the manuscript for clarity.
1. The timing of data collection would be important to note (was this prior to the onset of COVID procedures for giving birth in hospitals in Italy or after?)
2. Demographic information about the mothers would also be helpful for context (i.e. how many were married/living with a partner or single? Socioeconomic status or ability to financially provide for the new baby?). All of these variables are also really important to consider when looking at maternal stress so without that context some of the results lose their meaning and could be beneficial to strongly connect to the literature in your introduction
3. I'm unclear if you are assuming that all of the mother's are first time mothers, or that you have that confirmed in your data?
Reviewer 2 Report
The article entitled “Parenting stress, maternal self-efficacy and confidence in caretaking in a sample of mothers with newborns (0-1 month)” presents interesting results that may have application significance.
My comments below relate mainly to methodological issues:
1. how the effect and sample size were calculated? please provide the detailed information
2. where exactly the data were collected?
3. What were the inclusion criteria? was every woman invited to participate in the study during a routine visit? how many women were initially invited to participate, and what were the response rate and attrition rates? There is no detailed description of the test procedure - did participants fill in these questionnaires during the medical visit? did they receive the questionnaires at home and hand them back on the next visit? Who explained the study's purpose, course and conditions - the doctor during the visit or some research assistant?
4. The authors compared the obtained results with the results available in the literature. To correctly interpret the results, it would be good to know the sample size in the previous study and whether the conditions allow the use of a parametric test
5. Apart from the age of the respondents, the authors do not provide any sociodemographic and/or gynaecological and obstetric variables, therefore it is difficult to assess whether the obtained results cannot be determined in any way by other variables.
Introducing the suggested additions may be important for the interpretation of the results. If so, I ask the authors to include it in the discussion.
Round 2
Reviewer 2 Report
Thank you for providing changes and additional information. I am satisfied.